# Molecular Stratification of Childhood Ependymomas as a Basis for Personalized Diagnostics and Treatment

**DOI:** 10.3390/cancers13194954

**Published:** 2021-10-01

**Authors:** Margarita Zaytseva, Ludmila Papusha, Galina Novichkova, Alexander Druy

**Affiliations:** 1Dmitry Rogachev National Medical Research Center of Pediatric Hematology, Oncology and Immunology, 117997 Moscow, Russia; ludmila.mur@mail.ru (L.P.); Galina.Novichkova@fccho-moscow.ru (G.N.); dr-drui@yandex.ru (A.D.); 2Research Institute of Medical Cell Technologies, 620026 Yekaterinburg, Russia

**Keywords:** ependymoma, risk stratification, molecular group, prognosis

## Abstract

**Simple Summary:**

The current trend in neuropathology directs to the integrated histo-molecular approach. The traditional concept of histological grade should be complemented by comprehensive diagnostics with the mandatory use of molecular genetic markers. As a consequence, basic types of CNS tumors fall into multiple nosological entities that can be morphologically similar while having fundamentally different pathogenesis and clinical presentation. This trend is particularly evident for ependymal tumors, which harbor molecular markers of decisive importance for the prognosis. This minireview emphasizes recent achievements in ependymoma biology research closely connected with state-of-the-art diagnostics.

**Abstract:**

Ependymomas are among the most enigmatic tumors of the central nervous system, posing enormous challenges for pathologists and clinicians. Despite the efforts made, the treatment options are still limited to surgical resection and radiation therapy, while none of conventional chemotherapies is beneficial. While being histologically similar, ependymomas show considerable clinical and molecular diversity. Their histopathological evaluation alone is not sufficient for reliable diagnostics, prognosis, and choice of treatment strategy. The importance of integrated diagnosis for ependymomas is underscored in the recommendations of Consortium to Inform Molecular and Practical Approaches to CNS Tumor Taxonomy. These updated recommendations were adopted and implemented by WHO experts. This minireview highlights recent advances in comprehensive molecular-genetic characterization of ependymomas. Strong emphasis is made on the use of molecular approaches for verification and specification of histological diagnoses, as well as identification of prognostic markers for ependymomas in children.

## 1. Introduction

Ependymal tumors (ependymomas, EPNs), a common type of malignant neoplasms of the central nervous system (CNS), constitute about 10% of all intracranial tumors and about 20% of spinal cord tumors. EPNs rank third in the prevalence of pediatric CNS tumors (after glial and embryonal tumors) [1]. Despite the use of advanced protocols that include maximal safe surgical resection followed by localized radiotherapy, the mortality remains high due to frequent relapses explained by the strong metastatic potential of EPNs complemented by an efficient spread of metastases with cerebrospinal fluid.

Adverse predictors for EPNs are early age at onset, residual tumor tissue after resection, and metastatic lesions in CNS [2,3,4,5]; however, the detailed prognosis for EPNs is often hampered by (1) clinical and morphological diversity of the tumors and (2) complex relations of histopathological grades with the prognosis [6,7]. In line with the modern trends in neuropathology, the aggressiveness of a tumor and, accordingly, the prognosis is mainly determined by molecular-genetic aberrations, whereas the conventional, histologically defined grade becomes subsidiary [8,9]. For pediatric EPNs, the relevance of molecular stratification is especially obvious.

Over recent years, the diagnostics of CNS malignancies has been significantly reconsidered. The accent has been shifted from pathomorphology to molecular profiling and the search for clinically informative markers that would justify the selection of a particular therapy. Molecular framework-based stratification schemes have been developed and introduced into clinical practice for a number of CNS tumors; examples include *IDH1/2* mutations and 1p/19q codeletions for gliomas and oligodendrogliomas [10]; *KIAA1549-BRAF* fusions, *MYB*/*MYBL* rearrangements, recurrent pathogenic mutations in *BRAF* and *H3F3A* for pediatric astrocytomas [11,12]; and four molecular groups with the account of *MYC*/*MYCN* amplification for medulloblastomas [13].

EPNs of different molecular etiologies occupy distinct anatomical compartments within CNS. Recurrent genetic or epigenetic alterations found in EPNs are invariably linked to tumor localization. Molecular subgrouping of EPNs is superior to histopathological grading based on the WHO criteria [14]. Gene expression signatures and related subgrouping have shown the highest prognostic value among other studied molecular criteria. A tumor retains its affiliation to a particular subgroup indefinitely (it cannot be switched during progression and/or relapse of the disease), which increases its clinical significance [5,14,15]. An advanced EPN classification has been recently proposed by the Consortium to Inform Molecular and Practical Approaches to CNS Tumor Taxonomy (cIMPACT-NOW) update 7, aimed at connecting localization-dependent molecular groups with tumor progression modes and outcomes [16]. This view has been supported by WHO experts and reflected in the summary of the upcoming fifth edition of the WHO Classification of Tumors of the Central Nervous System (WHO CNS5) [9]. According to the newest CNS tumor nomenclature, ependymomas are subdivided into supratentorial (ST-EPNs), infratentorial (a.k.a. posterior fossa ependymomas, PF-EPNs), and spinal (Sp-EPNs) by localization of the primary tumor; these groups are further stratified by (epi)genetic features.

## 2. Molecular Profiles of ST-EPNs

ST-EPNs are fairly rare and show considerable genetic heterogeneity. ST-EPNs have been recently stratified into two major groups: supratentorial ependymoma, *ZFTA* fusion-positive (ST-EPN-ZFTA) and supratentorial ependymoma, *YAP1* fusion-positive (ST-EPN-YAP1) [9] consistently with gene expression and/or DNA methylation signatures revealed by transcriptomic methods and/or whole-genome DNA methylation profiling, respectively.

### 2.1. ST-EPN-ZFTA Group

Gain-of-function rearrangements in *ZFTA* or *YAP1* are specific for ST-EPNs. At that, ST-EPN-ZFTA tumors are prevalent (50–75% and 25% of ST-EPNs in children and adults, respectively [7,14,17,18,19,20]), while ST-EPN-YAP1 tumors are rare (3–10% in different cohorts [7,14,18,19,21,22]). The archetypal chimeric transcript harbored by *ZFTA*-rearranged ependymomas is *ZFTA*–*RELA*, hence the ST-EPN-RELA is a traditional designation for this group [14]. Alternative *ZFTA* fusions (non-*RELA*, e.g., *ZFTA-NCOA1*, *ZFTA-NCOA2*, *ZFTA-MAML2* [23,24,25,26,27,28], and *MN1-ZFTA* [28]) are less common.

Recurrent *ZFTA*–*RELA* fusion is a unique molecular hallmark of *ZFTA*-positive EPNs not found in other CNS tumors. Nine different transcript variants have been described, differing by breakpoints in *RELA* and its partner gene; the prevalent isoform comprises *ZFTA* exon 2 spliced to *RELA* exon 2 [17,22,29]. Formation of the *ZFTA*–*RELA* intrachromosomal gene fusion results from multiple double-strand breaks in 11q13.1 with subsequent random reassociation (typical for chromotrypsis); hence the diversity of fusion points for such transcripts. The oncogenic impact of classical *ZFTA–RELA* fusions was elucidated in recent studies [30,31].

*RELA* encodes the RelA (p65) subunit of the dimeric nuclear factor-κB (NF-κB), most known as a master regulator of immune responses and inflammation. NF-κB promotes apoptosis inhibition, cell growth, and pro-angiogenic signaling—the basic components of oncogenesis and tumor progression. Expression of *RELA* chimeras results in constitutive activation of NF-κB signaling pathway [17] and associated resistance of the tumor to chemo and radiation therapies [32]. ZFTA chimeric proteins accumulate in the nucleus. A zinc finger domain in the truncated ZFTA protein endows the chimeras with extraordinary high affinity to DNA. The oncoprotein interferes with chromatin structure at ST-EPN-associated loci, enabling the RELA transactivation domain to induce their transcription [30]. Moreover, apart from the canonical NF-κB pathway activation, *ZFTA*–*RELA* fusions may trigger other gene expression programs through recruitment of transcriptional co-activators BRD4, EP300, and CBP, which participate in chromatin-related pathways and represent potential druggable targets [31,33].

EPNs with the *ZFTA* gene fused with a non-*RELA* partner gene are considerably less common. These tumors have variable histological structures and, apart from the ependymomal component, may additionally involve pleomorphic xanthoastrocytoma-like, astroblastoma-like, malignant teratoma-like, embryonal tumor-like, or sarcoma-like patterns. Despite the heterogeneous morphology, these tumors are (epi)genetically similar and tend to resemble the classic *ZFTA*–*RELA*-fused EPNs, as revealed by methylome assay. A detailed analysis of DNA methylation profiles allows subdivision of these tumors into two clusters, one of them comprising tumors with histological features of astroblastomas and xanthoastrocytomas, harboring *ZFTA*–*MAML2* and *MN1*–*ZFTA* rearrangements; the second cluster comprises tumors histologically resembling small-cell sarcomatoid carcinomas and undifferentiated sarcomas, harboring *ZFTA*–*NCOA1* and *ZFTA*–*NCOA2* rearrangements [24,25,26,27,28].

The presence of recurrent *ZFTA*–*RELA* fusions has been repeatedly implicated as an adverse prognostic factor [5,14,18]. Five-year rates of event-free survival (EFS) and overall survival (OS) for ST-EPN-ZFTA tumors never exceed 29% and 75%, respectively [14]. Within the ST-EPN-ZFTA group, additional risks of relapse have been associated with 1q gains [5]. Interestingly, the St Jude Young Children 07 (SJYC07) study (encompassing ependymal tumors diagnosed in <3-year-olds) identified similar 4-year EFS rates for ST-EPN-ZFTA, ST-EPN-YAP1, and PF-EPN group A [7]. Consistently, two other studies conducted independently by the Italian Association of Pediatric Hematology and Oncology AIEOP and the Children’s Oncology Group (trial ACNS0121) revealed no difference in survival rates for ST-EPNs with and without *ZFTA*–*RELA* fusion [4,34].

*ZFTA* (non-*RELA*)-fused EPNs have an especially dismal prognosis, with EFS rates significantly lower compared with classical *ZFTA*–*RELA*-fused EPNs, while the corresponding OS rates are comparable [26]. However, these findings are preliminary, given the small number of cases reported so far. In the context of ST-EPN heterogeneity, it might be useful to consider *ZFTA* fusions with atypical (non-*RELA*) partners as a distinguishing feature for a separate group, the prognostic and clinical relevance of which is yet to be specified.

### 2.2. ST-EPN-YAP1 Group

ST-EPN-YAP1 tumors show an aberrant activity of transcription co-activator YAP1 (Yes-associated protein 1) related to its abnormal accumulation in the nucleus. With YAP1 being a direct regulator of TEAD and SMAD transcription factors, its escape from Hippo-dependent sequestration through accumulation in the nucleus results in sustained proliferative signaling via WNT and Hedgehog. More accurately, the nucleus accumulates the oncogenic fusion protein YAP1–MAMLD1 transferred from the cytoplasm to the nucleus independently of its YAP1-Ser127 phosphorylation status that limits the nuclear import of YAP1 in normal cells [35]. Apart from the prevalent *YAP1*–*MAMLD1* fusions, ST-EPN-YAP1 may harbor relatively rare structural variants, e.g., *YAP1*–*FAM118B* [14,36]. In some cases, the formation of *YAP1* fusions involves focal copy number alterations mapping to the 3′ portion of the gene (11q22.1–11q21.2) [37]. Unlike *ZFTA*-positive ependymomas, ST-EPN-YAP1 tumors have balanced genomes with local aberrations in the *YAP1* locus and no evidence of chromothripsis.

Andreiuolo et al. (2019) reported a multicenter retrospective study on what is so far the largest cohort of patients with *YAP1*-positive EPNs (*n* = 14). Overall survival for these patients (median observation time of 4.8 years within the range of 0.6–16 years) constituted 100%. It is important to note that the boy-to-girl ratio for the studied cohort was 1:6.5, and only three of the patients were over three years old at the time of diagnosis (the median age at diagnosis constituted 8.2 months) [37]. The best survival rates for the ST-EPN-YAP1 group among other EPNs were also reported by other authors [7,14]. Careful de-escalation of conventional EPN treatment protocols specifically for ST-EPN-YAP1 patients is currently under scrutiny. An opportunity to exclude (delay or dismiss) radiation therapy alleviates the risks of severe cognitive dysfunctions, endocrinopathies, and secondary tumors [38].

### 2.3. Non-ZFTA/Non-YAP1 ST-EPNs

The molecular diversity of ST-EPNs exceeds the currently established ZFTA-YAP stereotype. Tumors with neither *ZFTA* nor *YAP1* alterations are considered as a separate group, and recent findings emphasize the need for a finer specification. A distinct entity is formed by *PLAGL1* rearranged EPNs, harboring *EWSR1*–*PLAGL1* and less commonly *PLAGL1*–*FOXO1* or *PLAGL1*–*EP300* fusions [39], which echoes molecular landscapes of soft tissue sarcomas and a group of rare mesenchymal (non-meningothelial) and glioneuronal CNS tumors with *EWSR1*–non-ETS fusions [40,41]. Nevertheless, for the vast majority of ST-EPNs lacking recurrent chromosomal rearrangements, the oncogenic driver events remain elusive. Several reports reveal the presence of fusion genes *MAML2*–*ASCL2*, *MARK2*–*ADCY3* [19], *PTEN*–*TAS2R1* [14], *PATZ1*–*MN1*, *MYH9*–*SEC14L2*, *MTMR3*–*NCOA3* [24], *TMEFF2*–*FOXO1*, *PCGF1*–*CREBBP* [20], *FOXO1*–*STK24*, as well as *EP300*–*BCORL1* in such tumors [21]. Olsen et al. (2015) described two cases of hemispheric infantile EPN-like gliomas with *ALK* fusions (*CCDC88A*–*ALK* and *KTN1*–*ALK*), both of them morphologically ambiguous: the tumors showed glial phenotypes and resembled glioblastomas [42]. In the summary of the upcoming WHO CNS5, such tumors have been reclassified and renamed as infant-type hemispheric gliomas harboring receptor tyrosine kinase gene rearrangements [9]. Torre et al. (2020) reported in-frame fusions *AGK*–*BRAF* and *MYO5A*–*NTRK3* as potential targets for therapeutic inhibition [20]. In sum, these observations indicate the absence of a single driver mechanism for this group of tumors while underscoring the importance of their distinction from other CNS neoplasms.

## 3. Molecular Profiles of PF-EPNs

PF-EPNs, more prevalent in children than adults, constitute about 2/3 of intracranial ependymal tumors of childhood. Based on high-throughput molecular techniques, PF-EPNs are subdivided into two molecular groups: PF-EPN group A (PF-EPN-A), and PF-EPN group B (PF-EPN-B) [9].

### 3.1. PF-EPN-A Group

This highly heterogeneous group comprises 85–90% of infratentorial EPNs. PF-EPN-A tumors are often located laterally within the posterior fossa and occur predominantly in infants and young children, twice more frequently in boys than in girls; the average age at diagnosis constitutes 3.5 years [14]. The patients are at high risk of relapse, even under multimodal therapy and in the absence of extra adverse prognostic markers [5]. Identification with PF-EPN-A represents a strong independent prognostic factor associated with the worst rates of survival. According to Zapotocky et al. (2019), 5-year and 10-year EFS for PF-EPN-A constitute, respectively, 43% and 37% [3]. Relapses are typical for PF-EPN-A but not PF-EPN-B and, consistently, 10-year OS rates for PF-EPN-A are significantly lower than for PF-EPN-B (56–58% vs. 88–100%) [4,14,15]. Additional negative clinical predictors for PF-EPN-A are the presence of residual tumor tissue (incomplete resection) and adjuvant radiation therapy refusal [3]. Retrospective evaluation of outcomes for the patients receiving treatment under HIT-2000 protocol implicated residual tumors, 1q gains, and high mitotic activity of tumor cells (>10 mitotic figures per 10 fields of view) as independent adverse predictors for PF-EPNs in general and PF-EPN-A tumors in particular [2]. Cytogenetic prognostic factors for PF-EPNs include 1q gains and 6q losses [43,44]. These cytogenetic abnormalities are detected in 18.9% and 8.6% of PF-EPNs, respectively. At that, the 1q gained PF-EPNs harbor 6q losses at an increased frequency of 17.7% [43]. Both types of copy number variations have been qualified as adverse predictors. Five-year progression-free survival rates were 50% for EPNs without 1q gain and 6q loss, as opposed to 32% for 1q gain only, 7.3% for 6q loss only, and 0% for both 1q gain/6q loss tumors [43]. The ultra-high risks conferred by the co-occurrence of cytogenetic markers in PF-EPN-A patients should be taken into account for the treatment regimen optimization.

PF-EPN-A tumors reveal characteristic aberrant patterns of DNA methylation, the so-called CpG-island methylator phenotype (CIMP) with extensive hypermethylation of CpG-islands in promoter regions of multiple genes. This effect critically interferes with the function of PRC2 (Polycomb repressive complex 2) [45,46]—a transcription repressor protein complex that facilitates methylation of nucleosome histone H3 at amino acid residues H3K27 and H3K9 thus inhibiting the expression of key regulatory genes responsible for cell fate determination and differentiation. Bayliss et al. (2016) revealed the deficiency or complete loss of H3K27me3 in PF-EPN-A tumors [46]. This finding complements the earlier hypotheses on the central role of epigenetic mechanisms in PF-EPN pathogenesis inferred from the absence of presumably pathogenic mutations in chromatin remodeling genes and enzymes that catalyze post-translational modifications (e.g., methylation) of histones in whole-genome sequencing data [17,45,46]. The recruitment of Polycomb group (PcG) transcription repressors to chromatin requires the presence of non-methylated CpGs; accordingly, the loss of H3K27me3 methylation has been associated with dense hypermethylation of CpG-islands preventing the recruitment of PcG proteins to chromatin by steric hindrance.

An advanced investigation of molecular mechanisms responsible for the observed epigenetic malfunctioning revealed a plausible association of the H3K27me3 deficiency with elevated expression levels of accessory proteins encoded by *EZHIP* (formerly *CXorf67*) and *EPOP* (formerly *C17orf96*) [47,48,49]. As demonstrated by Hübner et al. (2019), EZHIP is a competitive inhibitor of PRC2. A conservative stretch of amino acids in the C-terminal portion of EZHIP mimics the K27 methylation target in histone H3, albeit with K27M substitution. The binding of methionine M27 (instead of lysine K27) to the active center in the histone-lysine N-methyltransferase subunit of PRC2 blocks its catalytic activity [50]. Somatic missense mutations in *EZHIP* are detected in a small proportion of PF-EPN-A tumors (<10%) [48]. Jain et al. (2019) demonstrated that such mutations have no influence on H3K27me3 levels thus disproving their functional significance [51]. Noteworthy, no loss-of-function mutations in *EZHIP* (nonsense substitutions or frameshift indels) have been reported. Elevated expression of *EZHIP* in tumors may be caused by mutations in *cis*-regulatory elements; the same effect may be conferred by the formation of fusion genes involving *EZHIP* locus (for instance, *MBTD1*–*EZHIP* fusion described for low-grade endometrial stromal sarcoma [52]). However, no fusions comprising *EZHIP* or PRC2 subunit-encoding genes (e.g., *EED*, *SUZ12*) have been described for EPNs.

Related signatures of disrupted epigenetic regulation have been associated with *H3* K27M mutations typical for diffuse midline gliomas (DMGs) but rarely found in PF-EPN-A (<5% of the cases). Noteworthy, in EPNs such mutations are harbored by canonical histone-encoding genes *HIST1H3C* and *HIST1H3B*, whereas in DMGs they are predominantly found in a replacement histone gene *H3F3A* (90% of the cases) [48,53,54,55]. Given the mutually agonistic roles of the onco-histone H3 K27M and EZHIP, it would be natural to expect similar patterns of disease progression and therapy outcomes for *H3* K27M-mutant DMG and EZHIP^high^ PF-EPN-A. Indeed, in DMG, disruption of H3K27me3-mediated epigenetic regulation is associated with an extremely aggressive course of the disease, typically presenting with sustained tumor growth and polychemotherapy resistance [56,57,58]. Similarly, effective chemotherapy regimens for PF-EPN-A are missing [59] and therapeutic options for relapses are extremely limited [5,60,61,62,63].

Despite the uniformity of methylation profiles within PF-EPN-A, tumors of this group show considerable molecular heterogeneity and can be additionally classified into two major subgroups A1 and A2 (and ultimately into nine minor subtypes by using additional markers: gains 1q, deletions 22q, 6q, and 10q, and OTX2 protein expression). PF-EPN-A1 tumors are distinguished by pronounced expression of the homeotic HOX genes (*HOXA1/2/3/4*, *HOXB2/3/4*, *HOXC4*, and *HOXD4*) which define the segmental (rhombomeric) organization of the hindbrain in early embryogenesis. PF-EPN-A2 tumors hyperexpress *EN2*, *CNPY1*, and *IRX3*—a group of genes involved in the rhombomere differentiation. Expression of A1- and A2-specific genes within the developing hindbrain shows distinct zonality—increased expression of A2 markers is characteristic of the rostral portion at the border with the midbrain, while expression of HOX genes is more pronounced in caudal segments of the brainstem and spinal cord. Differential expression of the spatial patterning genes in A1 and A2 tumors apparently reflects their origin from different hindbrain structures. However, the practical relevance of the advanced A1/A2 subgrouping is questionable. Stratification by clinical factors (gender, age at diagnosis, tumor resection volume, and received therapy) revealed no significant differences between A1 and A2 tumors, except the patterns of relapse (PF-EPN-A1 tumors more often produce local than distant relapses, and vice versa) [48].

### 3.2. PF-EPN-B Group

In contrast to PF-EPN-A tumors which predominantly affect children, PF-EPN-B tumors are more common in adults. In adolescents (aged 10–17), about 45% of newly diagnosed EPNs fall into this group. The prognosis for PF-EPN-B tumors is favorable: 10-year OS rates for the patients after subtotal and gross total resections reach 66.7% and 96.1%, respectively [3,14,15,64]. Thus, the prognosis for this group strongly depends on the extent of surgical resection. The occurrence of delayed relapses (10 years after the onset) underscores the importance of long-term follow-up [64]. Patients with R0 may benefit from chemo- and radiation-sparing strategies; such possibility is being considered [38]. The observed difference in patterns of recurrence between PF-EPN-A and PF-EPN-B adds to the relevance of comprehensive molecular characterization of a tumor as early as possible.

By now, recurrent mutations or fusion genes in PF-EPN-B tumors are missing, and no clear drivers for this group have been identified. Ciliogenesis and microtubule assembly are deregulated only in PF-EPN-B tumors, while several canonical cancer-associated pathways operate in the PF-EPN-A group (VEGF, PDGF, EGFR, RAS signaling, etc.) [14]. PF-EPN-B tumors harbor major cytogenetic aberrations including gains 1q, monosomies 6, 10, and 17, trisomies 5, 8, and 18, and deletions 22q [64]. The diversity of cytogenetic profiles revealed for PF-EPN-B indicate inherent genomic instability and suggest that these tumors emerge from multiple driving events. Similarly with PF-EPN-A, the PF-EPN-B group shows significant heterogeneity, with distinct molecular subtypes of different demographics, copy number alterations, and gene expression signatures. By contrast with PF-EPN-A, gains 1q pose no extra risks for PF-EPN-B tumors. Losses 13q may represent a more reliable negative prognostic marker than gains 1q; however, this assumption requires further substantiation, particularly as the basis for de-escalation of therapy regimens. However, the extent of resection remains the strongest predictor of poor outcomes for this group. Given the patient data scarcity, advanced stratification within PF-EPN-B remains clinically irrelevant [64].

### 3.3. ST-EPN-ZFTA-like PF-EPNs

Unique cases of *ZFTA*–*MAML2*, *ZFTA*–*RELA*, and *ZFTA*–*NCOA2* fusion in PF-EPN were reported recently. These tumors revealed characteristic *ZFTA*-mediated gene expression and whole-genome DNA methylation signatures corresponding to the ST-EPN-ZFTA group; accordingly, they were classified as “ST-EPN-ZFTA” despite the infratentorial localization [65].

A summary of the intracranial EPN classification is given in Figure 1.

## 4. Molecular Profiles of Sp-EPNs

Sp-EPNs constitute a heterogeneous group with a generally favorable prognosis. These tumors mostly occur in adult patients and are rare in children. Clinical outcomes for Sp-EPNs are better than for intracranial EPNs, with 5-year OS rates within the range of 60–90% [14]. Three molecular groups of Sp-EPNs were originally identified, including subependymomas, myxopapillary Sp-EPNs (SP-MPE), and Sp-EPNs per se; the molecular subgrouping shows excellent concordance with corresponding histopathological subtypes [14]. Our knowledge on the molecular pathogenesis of Sp-EPN tumors is limited. The groups reveal characteristic somatic copy number aberrations; most Sp-EPNs harbor 22q deletions involving neurofibromin 2 (*NF2*) tumor suppressor gene, whereas SP-MPEs show chromosomal instability.

SP-MPEs, the most prevalent type of pediatric spinal cord EPNs, predominantly arise in the conus medullaris, cauda equina, and filum terminale regions [66]. Despite their low mitotic index and slow-growing nature, SP-MPEs generally have far more aggressive behavior than other low-grade CNS tumors. Furthermore, pediatric SP-MPEs are especially aggressive, with much higher rates of local recurrence and secondary seeding to distant craniospinal sites or local spinal sites (64% cf. 32% in adults) [67]. As demonstrated by Ahmad et al. (2021), pediatric SP-MPEs exhibit aberrant activity of the mitochondrial metabolic pathways [68]. The only recurrent focal amplification identified for this group involves *HOXB* gene cluster mapping to 17q. *HOXB13* amplification represents a candidate diagnostic marker for SP-MPEs. The elevated expression of *HOXB13* enhances tumor cell proliferation and dissemination, playing a critical role in the development of metastasis [68]. Due to the high propensity for local recurrence and distant neural axis dissemination, the summary of the upcoming WHO CNS5 identifies SP-MPEs with grade 2 (rather than grade 1). SP-MPEs have distinctive histopathological features such as well-organized papillary architecture, with vascular cores and abundant mucinous extracellular matrix. Histological examination of tumor tissue is necessary and sufficient for the diagnosis of SP-MPE, whereas genetic testing is accessory [9].

A rare subtype of Sp-EPN in adult patients has been described recently, presenting highly aggressive clinical behavior with early metastasis, diffuse leptomeningeal spread throughout CNS, and resistance to standard treatment protocols. All of them harbored *MYCN* amplification and no other recurrent pathogenic events [69,70,71]. Importantly, these tumors formed a distinct methylation cluster of their own, and none of them clustered with any of the previously identified nine EPN groups. Recognizing the importance of clinical and molecular data on such tumors, the summary of the upcoming WHO CNS5 reports a novel nosological entity of *MYCN*-driven Sp-EPNs with dismal outcomes [9].

Genetic alterations found in particular in EPN groups are summarized in Table 1. Subgroup-specific diagnostic and candidate genes in pediatric EPNs are contained in Table 2.

## 5. Molecular Profiles of Subependymomas

Subependymomas are exceptionally rare slow-growing benign neoplasms, accounting for only 0.07–0.7% of all intracranial tumors [14,72]. These tumors typically arise in adults (aged 22–76 years) without strict predilection to a particular site in CNS. The most typical locations, the fourth and lateral ventricles, are encountered in up to 85% of the cases, followed by septum pellucidum, brainstem, and spine [73]. DNA methylation profiles of subependymomas differ depending on tumor localization, which provides certain grounds for molecular subgrouping [14]. However, all subependymomas have a favorable prognosis independently of localization. According to the cIMPACT-NOW group recommendations and the summary of the upcoming WHO CNS5, morphological examination provides adequate means for the diagnosis and prognostication; the integrative histo-molecular approach for subependymomas is accessory [9,16]. Recent findings suggest that the process of clonal evolution in subependymomas may give rise to more aggressive tumor clones enriched with pure EPN phenotypes, chromosome 6 losses, and *TERT* mutations. These markers, associated with increased risks of recurrence, should be considered as an indication for more intensive therapies, especially under conditions of subtotal tumor resection [44].

## 6. Laboratory Approaches for EPN Diagnostics

From a histological perspective, EPNs show moderate cellularity and variable mitotic activity; they consist of monomorphic rounded or oval cells with scant cytoplasm and vesicular nuclei containing granular (salt-and-pepper) chromatin. Key histological features of EPNs are perivascular (pseudo) rosettes and ependymal rosettes. EPNs are positive for glial fibrillary acidic protein (GFAP), S100, vimentin, rarely cytokeratin, and epithelial membrane antigen (EMA, positive along the luminal surface of ependymal rosettes or as characteristic dot-like or ring-like intracytoplasmic patterns) and negative for most neuronal antigens [6].

No unified standards for molecular diagnostics of EPNs have been introduced so far. The molecular group for each particular case is determined by whole-genome DNA methylation profiling as a golden standard. Comprehensive analysis of DNA methylation signatures is the method of choice for differential diagnosis within a broad spectrum of glial and embryonal tumors including CNS neuroblastoma with *FOXR2* activation, CNS Ewing sarcoma family tumor with *CIC* alteration, CNS high-grade neuroepithelial tumor with *MN1* alteration, and CNS high-grade neuroepithelial tumor with *BCOR* alteration [24,74]. The advantage of using this technique is the suitability of formalin-fixed paraffin-embedded tumor tissue for the analysis (methylated sites in genomic DNA are preserved during fixation, dehydration, etc.). However, the use of DNA methylation assays in routine laboratory practice has serious limitations, as the data processing is complex and the method itself is expensive, sophisticated, and labor-consuming. Alternative diagnostic algorithms may be based on a combination of economically justified methods including histological examination, immunohistochemistry (IHC) testing, reverse transcription-polymerase chain reaction (RT-PCR), Sanger sequencing, fluorescence in situ hybridization (FISH), and probably also the NanoString nCounter^®^ platform (Figure 2).

### 6.1. Differential Diagnosis of ST-EPNs

In many cases, unambiguous determination of the molecular group of a tumor can be afforded by a certain combination of routine morphological and genetic approaches [75]. For instance, nuclear accumulation of p65 (RelA) and the presence of *ZFTA*–*RELA* fusion are sufficient for the identification of the sample with the core ST-EPN-ZFTA group harboring the classical *ZFTA*–*RELA* transcript. The diversity of fusion transcripts, which results from the variability of breakpoints in *RELA* and its partner genes, significantly complicates the identification of particular rearrangements and their use as markers. Conventional RT-PCR tests are targeted at the two most frequent fusions *ZFTA* (exon 2)–*RELA* (exon 2) and *ZFTA* (exon 3)–*RELA* (exon 2) [19,75,76].

A highly efficient way of screening for *ZFTA* and *RELA* rearrangements is provided by FISH with break-apart probes for one of the fusion partner genes. This approach allows detecting rearrangement of the gene of interest without accurate determination of the breakpoint [19,21]. Its excellent concordance with other methods including nuclear expression of RelA and DNA methylation profiling should be noted [19].

*ZFTA* gene rearrangements induce the hyperexpression of L1 cell adhesion molecule (L1CAM) and/or cyclin D1 may be considered as a surrogate marker of the ST-EPN-ZFTA group [18,20,22,23]. However, IHC is not sufficient on its own: the results require confirmation by independent alternative methods since neither of the antibodies (anti-p65, anti-L1CAM, anti-cyclin D1) has enough sensitivity and/or specificity to reliably verify the ST-EPN-ZFTA group [20,22]. The diagnosis of ST-EPN-ZFTA with alternative (non-*RELA*) fusion gene can be suspected if tumor cells show immunopositivity for L1CAM and negativity for p65 [26].

Identification of a tumor with the ST-EPN-YAP1 group is based on detection of either *YAP1* rearrangement by FISH or one of the *YAP1*-fusions (*YAP1*–*MAMLD1*, *YAP1*–*FAM118B*) by RT-PCR, while IHC with anti-YAP1 and/or anti-Claudin-1 is noninformative [19,37].

### 6.2. Differential Diagnosis of PF-EPNs

Panwalkar et al. (2017) proposed IHC tests for H3K27me3 as a straightforward and affordable method for PF-EPN stratification. The authors demonstrate that PF-EPN-B tumors are completely positive for H3K27me3, whereas the presence of H3K27me3 signal in less than 80% of tumor cell nuclei (against the total positivity of endothelial cells used as an internal control) is indicative of PF-EPN-A. Moreover, groupings based on whole-genome DNA methylation profiling vs. IHC tests for H3K27me3 were 99.1% concordant; the outlier was H3K27me3-negative tumor from a 12-year-old female patient, classified as PF-EPN-B on the basis of DNA methylation analysis [77]. The methodology was further enhanced by Fukuoka et al. (2018) who attempted at finer quantitation of the IHC data. In about 62% of prediagnosed PF-EPN-A tumors examined by the authors, the H3K27me3 reactivity was shown by 5–50% of the tumor cell nuclei; for the rest of PF-EPN-A tumors (about 40%), the proportion was less than 5%. In contrast, the vast majority of PF-EPN-B tumors showed intact expression of H3K27me3, with 90–95% of tumor cell nuclei stained positive, except two cases of reduced positivity (10–60%) classified as PF-EPN-B by DNA methylation profiling. Overall, the cut-off level of 80% H3K27me3-positive tumor cells allowed distinguishing PF-EPN-B from PF-EPN-A with 100% specificity and 86.7% sensitivity [21].

Antin et al. (2020) suggested complementing the routine IHC panels for CNS tumor diagnostics with anti-EZHIP staining. The intensive diffuse nuclear staining with >90% tumor cells immunopositive was obtained for PF-EPN-A (with the exception of rare *H3* K27M-mutant tumors), diffuse midline gliomas with wild-type (non-mutant) *H3F3A*, and germinomas [78]. Nambirajan et al. (2021) evaluated an extended panel of IHC markers (H3K27me3, acetyl-H3K27, H3K27M, ATRX, EZHIP, EPOP, and Tenascin-C) for the use in differential diagnostics of PF-EPNs. The authors demonstrated that a combination of EZHIP- and H3K27me3-specific antibodies is sufficient for reliable verification of PF-EPN-A; this finding illustrates the consistency between genetic and IHC profiles of a tumor [79].

Given the existence of rare EPN subtypes e.g., PF-EPN with *ZFTA* rearrangement [65] or ST-EPN harboring *ZFTA* fusions with atypical (non-*RELA*) partners, correct identification of a tumor with a particular group requires an extended diagnostic algorithm accounting for gene expression signatures. The creation of a universal diagnostic tool for the determination of molecular subtypes of EPN can be based on the analysis of gene expression signatures using the NanoString nCounter^®^ platform. The assay must be carried out with customized panels of markers, corresponding to EPN group-specific signatures of upregulated genes, designed specifically for this purpose. Lastowska et al. (2021) identified group-specific marker genes for *RELA*-fused EPN (*RELA*, *ELL3*, *FBP2*, *PCP4L1*, and *MYO3A*), *YAP1*-fused EPN (*MRAP*, *IGF1*, *CAPS*, and *WWC1*), PF-EPN-A (*LAMA2*, *ALDH1L1*, *SLC6A13*, *IGSF1*, and *CXorf67*), and PF-EPN-B (*NELL2*, *DNAH1*, *CEP83*, *C9orf72*, and *NXNL2*) tumors. The NanoString-based approach clearly separated PF-EPNs into two clusters, corresponding to the defined PF-EPN groups, based on the expression of selected genes. Extended cluster analysis allowed subdivision of PF-EPN-A tumors into PF-EPN-A1 and PF-EPN-A2 subgroups using four marker genes (*SKAP2*, *WIF1*, and *EN2*, *CNPY1*, respectively). The described gene panel for *RELA*-positive EPN did not allow differentiation of EPN with *ZFTA*–*MAML2* fusion [80]. The potential of the NanoString method for diagnosis and advanced classification of other CNS tumors has been confirmed in several studies [81,82,83]. The approach provides reliable stratification of medulloblastomas into four groups (designated WNT, SHH, group 3, and group 4) based on the evaluation of 22 transcripts expressed differentially among the groups and contributing to characteristic group-specific gene expression signatures [81,82]. Primitive neuroectodermal tumors of the CNS (CNS-PNETs) have been already defined as four new molecularly defined entities (CNS neuroblastoma with *FOXR2* activation, CNS Ewing sarcoma family tumor with *CIC* alteration, CNS high-grade neuroepithelial tumor with *MN1* alteration, and CNS high-grade neuroepithelial tumor with *BCOR* alteration) using a single multi-gene tumor-specific signature [83]. However, differential diagnostics of EPNs in mixed series with other CNS tumors by the NanoString approach is impossible. For correct molecular stratification of ependymal tumors by the NanoString approach, all samples intended for comparison must be morphologically identified as EPN prior to the analysis.

## 7. Therapeutic Targeting of EPNs

Despite the profound insights into ependymal tumor biology, consensus recommendations for the management of patients with EPNs with regard to molecular diagnosis are missing. As tumors are resistant to conventional chemotherapy, the search for druggable targets is highly relevant. However, no candidate targets have been identified despite the extensive efforts in molecular profiling. Clinical studies on general cohorts showed no significant objective response to ERBB1/2 (lapatinib) and VEGF inhibitors (bevacizumab, sunitinib), despite the evidence on overexpression of ERBB2, ERBB4, or VEGF in EPNs [84,85,86,87].

In the molecular era, cancer-specific somatic aberrations should be taken into account when choosing a treatment strategy. For PF-EPNs, the pathogenesis of which involves epigenetic mechanisms, the possibility of using pharmaceuticals targeted at epigenetic modifications, including abnormalities of DNA methylation and histone modifications, seems most rational. Despite the promising results of pre-clinical research [88,89], histone deacetylase (HDAC) inhibitors showed no therapeutic activity in patients, apparently due to the intricacies of subcellular localization of HDACs and inability of the inhibitors to accumulate inside the brain tissue in concentrations sufficient for therapeutic response [90]. Nevertheless, the search for new HDAC inhibitors with appropriate brain-penetrating capacities and safety profiles may provide a useful treatment strategy [91]. The possibility of using BET-bromodomain inhibitors as anticancer therapeutics is being investigated in pediatric EPN stem cell models [92]. The upregulation of EZHIP could have important implications for therapeutic approaches. As reported by Han et al. (2020), EZHIP has a conservative PALB2-binding domain which enables its functioning as a competitive inhibitor of BRCA2. Elevated levels of EZHIP prevent the formation of BRCA1-PALB2-BRCA2 complexes thus inhibiting the homologous recombination-mediated DNA repair pathway. The results indicate the potential of PARP inhibitors as targeted therapeutics in PF-EPN-A, especially when combined with radiotherapy [61].

NF-kappa B inhibitors are considered potential therapeutic agents for the treatment of ST-EPN-ZFTA with constitutive activation of the NF-κB signaling pathway. A trial currently in phase II is evaluating the effectiveness of marizomib—a second-generation irreversible proteasome inhibitor, enrolling patients with histologically proven spinal or intracranial EPN, including but not limited to ST-EPN-ZFTA (ClinicalTrials.gov Identifier: NCT03727841, accessed on 20.09.2021). At the moment, the enrollment is complete, and the study is in progress. Specific expression of PD-L1 on both tumor and myeloid cells in ST-EPN-RELA has been demonstrated, accompanied by high levels of PD-1 expressed by tumor-infiltrating T cells (both CD4 and CD8) [93,94]. In the context of immunotherapy, ST-EPN-RELA progression may be controlled with PD-1 inhibitors, such as pembroluzimab or nivolumab [95].

Despite the principal shift in the EPN diagnostics and molecular stratification, its immediate impact on the existing treatment regimens is low. The correction would require preclinical and clinical trials for EPNs with due consideration of the molecular subgrouping.

## 8. Conclusions

As demonstrated by advanced studies of the last decade, ependymomas constitute a heterogeneous group of tumors and differ by molecular etiology. This minireview underscores the importance of comprehensive molecular profiling for ependymal tumors aimed at identifying specific expression signatures and/or (epi)genetic variants. Molecular identification of an ependymal tumor with a particular molecular group should follow its anatomical and histopathological assessment. Advanced stratification of patients into risk groups provides a framework for personalized management, e.g., allows de-escalation of the therapy in patients with low-risk tumors (supratentorial ependymomas group YAP1 and infratentorial ependymomas group B). Detailed understanding of causative molecular abnormalities for particular tumors is pivotal for the development of novel therapeutic options.

## Figures and Tables

**Figure 1 cancers-13-04954-f001:**
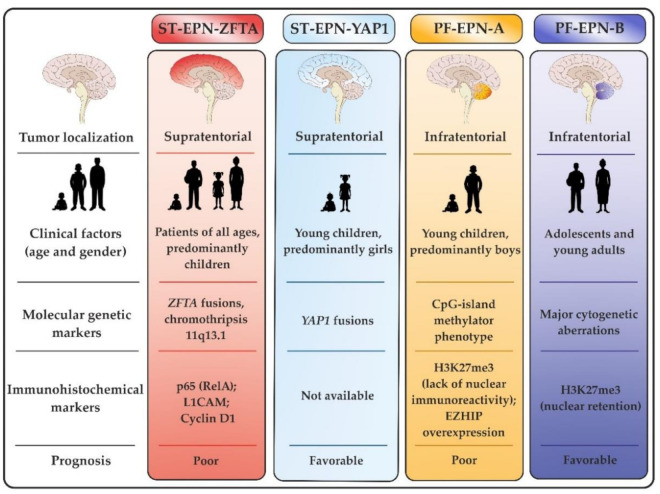
Basic classification of intracranial ependymal tumors.

**Figure 2 cancers-13-04954-f002:**
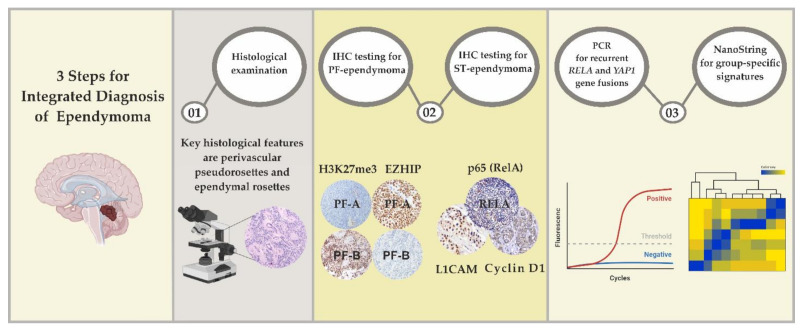
The integrated approach for molecular group determination applied to ependymal tumors.

**Table 1 cancers-13-04954-t001:** Summary of major molecular markers of ependymal tumors.

Localization	Molecular Group	Major Molecular Markers	Prevalence within the Group	Pathogenic Impact
Supratentorial	ST-EPN-ZFTA	*ZFTA*–*RELA* fusion, chromothripsis 11q13.1	90–95%	NF-kB pathway activation
		*ZFTA*–*MAML2* *ZFTA*–*NCOA1* *ZFTA*–*NCOA2*	5–10%	EP300/CREBBP gene expression pathway
	ST-EPN-YAP1	*YAP1*–*MAMLD1*	95%	Hippo pathway activation
		*YAP1*–*FAM118B*	5%	
Infratentorial	PF-EPN-A	*EZHIP* overexpression	95%	CpG-island methylator phenotype
		*HIST1H3C*, *HIST1H3B* or *H3F3A* K27M substitution	<5%	
	PF-EPN-B	Major cytogenetic aberrations	Up to 100%	Ciliogenesis deregulation
Spinal	Sp-MPE	HOXB cluster genes amplification	Up to 100%	Mitochondrial metabolism pathways activation
	SP-EPN-MYCN	*MYCN* amplification	100%	Proliferative signaling

**Table 2 cancers-13-04954-t002:** Subgroup-specific diagnostic marker and candidate genes involved in recurrent chromosomal abnormalities in pediatric EPNs.

Molecular Group	Implicated Gene *	Gene Name	Cytogenetic Band	Pathogenic Impact	Evidence Level **	Evidence-Based Categorization ***	Hallmark of Cancer ****	
							Promotes	Suppresses
ST-EPN-ZFTA	*ZFTA*	Zinc finger translocation associated	11q13.1	5′-partner gene in *ZFTA*–*RELA* fusion	I	Tier I, level A	Genome instability	
	*RELA*	V-Rel avian reticuloendotheliosis viral oncogene homolog A	11q13.1	3′-partner gene in *ZFTA*–*RELA* fusion	I	Tier I, level A	Escaping programmed cell death; tumor promoting inflammation	
	*MAML2*	Mastermind-like transcriptional coactivator 2	11q21	3′-partner gene in *ZFT–MAML2* fusion	III	Tier II, level C	Proliferative signaling; angiogenesis	
	*NCOA1*	Nuclear receptor coactivator 1	2p23.3	3′-partner gene in *ZFTA–NCOA1* fusion	III	Tier II, level C	Proliferative signaling; change of cellular energetics	
	*NCOA2*	Nuclear receptor coactivator 2	8q13.3	3′-partner gene in *ZFTA–NCOA2* fusion	III	Tier II, level C	Proliferative signaling; change of cellular energetics; escaping programmed cell death	
ST-EPN-YAP1	*YAP1*	Yes1-associated transcriptional regulator	11q22.1	5′-partner gene in *YAP1–MAMLD1* fusion	II	Tier I, level A	Proliferative signaling; escaping programmed cell death; invasion and metastasis	Escaping programmed cell death
	*MAMLD1*	Mastermind-like domain-containing 1	Xq28	3′-partner gene in *YAP1–MAMLD1* fusion	II	Tier I, level A	Proliferative signaling; angiogenesis	Escaping programmed cell death
	*FAM118B*	Family with sequence similarity 118 member B	11q24.2	3′-partner gene in *YAP1–FAM118B* fusion	IV	Tier II, level D	Unknown	
Non-ZFTA/Non-YAP1 ST-EPNs	*PLAGL1*	PLAG1-like zinc finger 1	6q24.2	3′-partner gene in *EWSR1-PLAGL1* fusion; 5′-partner gene in *PLAGL1–FOXO1* or *PLAGL1–EP300* fusion	IV	Tier II, level D	Suppression of growth	Escaping immunic response to cancer; tumor promoting inflammation; invasion and metastasis; angiogenesis
	*EWSR1*	EWS RNA binding protein 1	22q12.2	5′-partner gene in *EWSR1–PLAGL1* or *EWSR1–PATZ1* fusion	IV	Tier II, level D	Proliferative signaling; escaping programmed cell death; angiogenesis; invasion and metastasis	Genome instability and mutations
	*FOXO1*	Forkhead box O1	13q14.11	3′-partner gene in *PLAGL1–FOXO1* fusion	IV	Tier II, level D	Change of cellular energetics	Escaping programmed cell death
	*EP300*	E1A binding protein P300	22q13.2	3′-partner gene in *PLAGL1–EP300* fusion	IV	Tier II, level D	Suppression of growth	Escaping programmed cell death
	*PATZ1*	POZ/BTB and AT hook-containing zinc finger 1	22q12.2	3′-partner gene in *EWSR1–PATZ1* or *MN1–PATZ1* fusion	IV	Tier II, level D	Proliferative signaling; escaping programmed cell death	
	*MN1*	MN1 proto-oncogene, transcriptional regulator	22q12.1	5′-partner gene in *MN1-PATZ1* fusion	IV	Tier II, level D	Suppression of growth	Escaping programmed cell death
PF-EPN-A	*EZHIP*	EZH inhibitory protein	Xp11.22	Overexpression	IV	Tier II, level D		EZH1/EZH2-mediated trimethylation of H3K27
	*EPOP*	Elongin BC and polycomb repressive complex 2-associated protein	17q12	Overexpression	IV	Tier II, level D		EZH2-mediated trimethylation of H3K27
	*HIST1H3C*	H3 clustered histone 3	6p22.2	Somatic mutation	IV	Tier II, level D		EZH2-mediated trimethylation of H3K27
	*HIST1H3B*	H3 clustered histone 2	6p22.2	Somatic mutation	IV	Tier II, level D		EZH2-mediated trimethylation of H3K27
	*H3F3A*	H3.3 histone A	1q42.12	Somatic mutation	IV	Tier II, level D		EZH2-mediated trimethylation of H3K27
	*BCL9*	BCL9 transcription coactivator	1q21.2	Oncogene, involved in 1q gain	V	NA	Proliferative signaling; invasion and metastasis; angiogenesis	
	*ARNT*	Aryl hydrocarbon receptor nuclear translocator	1q21.3	Oncogene, involved in 1q gain	V	NA	Angiogenesis; change of cellular energetics	Invasion and metastasis
	*SETDB1*	SET domain bifurcated histone lysine methyltransferase 1	1q21.3	Oncogene, involved in 1q gain	V	NA	Epigenetic transcriptional repression by recruiting HP1 (CBX1, CBX3 and/or CBX5) proteins to methylated histones	
	*NTRK1*	Neurotrophic receptor tyrosine kinase 1	1q23.1	Oncogene, involved in 1q gain	V	NA	Proliferative signaling; escaping programmed cell death; angiogenesis	
	*FCRL4*	Fc receptor-like 4	1q23.1	Oncogene, involved in 1q gain	V	NA	Escaping immunic response to cancer	
	*FCGR2B*	Fc fragment of IgG receptor IIb	1q23.3	Oncogene, involved in 1q gain	V	NA	Suppression of growth	Escaping programmed cell death
	*DDR2*	Discoidin domain receptor tyrosine kinase 2	1q23.3	Oncogene, involved in 1q gain	V	NA	Invasion and metastasis	
	*PBX1*	PBX homeobox 1	1q23.3	Oncogene, involved in 1q gain	V	NA	Angiogenesis; escaping programmed cell death; change of cellular energetics	
	*ABL2*	ABL proto-oncogene 2, non-receptor tyrosine kinase	1q25.2	Oncogene, involved in 1q gain	V	NA	Proliferative signaling; invasion and metastasis; angiogenesis; genome instability and mutations; change of cellular energetics	Escaping programmed cell death
	*MDM4*	MDM4 regulator of P53	1q32.1	Oncogene, involved in 1q gain	V	NA	Proliferative signaling; invasion and metastasis; angiogenesis; escaping programmed cell death	Suppression of growth
	*ELK4*	ETS transcription factor ELK4	1q32.1	Oncogene, involved in 1q gain	V	NA	Proliferative signaling; escaping programmed cell death	
	*RGS7*	Regulator of G protein signaling 7	1q43	Oncogene, involved in 1q gain	V	NA	Change of cellular energetics	
	*AKT3*	AKT serine/threonine Kinase 3	1q43-q44	Oncogene, involved in 1q gain	V	NA	Proliferative signaling; suppression of growth; invasion and metastasis; angiogenesis; escaping programmed cell death; change of cellular energetics	Invasion and metastasis; angiogenesis; genome instability and mutations
	*EPHA7*	EPH receptor A7	6q16.1	Tumor suppressor gene, involved in 6q loss	V	NA		Escaping programmed cell death
	*CCNC*	Cyclin C	6q16.2	Tumor suppressor gene, involved in 6q loss	V	NA		Proliferative signaling
	*PRDM1*	PR/SET domain 1	6q21	Tumor suppressor gene, involved in 6q loss	V	NA	Suppression of growth	Escaping immunic response to cancer
	*FOXO3*	Forkhead box O3	6q21	Tumor suppressor gene, involved in 6q loss	V	NA	Change of cellular energetics	Escaping programmed cell death
	*PTPRK*	Protein tyrosine phosphatase receptor type K	6q22.33	Tumor suppressor gene, involved in 6q loss	V	NA	Escaping immunic response to cancer	Proliferative signaling
	*BCLAF1*	BCL2-associated transcription factor 1	6q23.3	Tumor suppressor gene, involved in 6q loss	V	NA		Escaping programmed cell death
	*TNFAIP3*	TNF alpha-induced protein 3	6q23.3	Tumor suppressor gene, involved in 6q loss	V	NA		Escaping immunic response to cancer; tumor promoting inflammation
	*LATS1*	Large tumor suppressor kinase 1	6q25.1	Tumor suppressor gene, involved in 6q loss	V	NA	Suppression of growth	Genome instability and mutations; escaping programmed cell death
	*ESR1*	Estrogen receptor 1	6q25.1	Tumor suppressor gene, involved in 6q loss	V	NA	Proliferative signaling; suppression of growth; escaping immunic response to cancer; invasion and metastasis	Invasion and metastasis
	*ARID1B*	AT-rich interaction domain 1B	6q25.3	Tumor suppressor gene, involved in 6q loss	V	NA	Suppression of growth; cell replicative immortality	Cell replicative immortality; invasion and metastasis; genome instability and mutations; escaping programmed cell death
	*QKI*	QKI, KH domain-containing RNA binding	6q26	Tumor suppressor gene, involved in 6q loss	V	NA	Suppression of growth; escaping programmed cell death	Escaping programmed cell death
PF-EPN-B	*LATS2*	Large tumor suppressor kinase 2	13q12.11	Tumor suppressor gene, involved in 13 q loss	V	NA	Suppression of growth; invasion and metastasis	Invasion and metastasis; genome instability and mutations; escaping programmed cell death
	*CDX2*	Caudal type homeobox 2	13q12.2	Tumor suppressor gene, involved in 13 q loss	V	NA		Proliferative signaling
	*BRCA2*	BRCA2 DNA repair associated	13q13.1	Tumor suppressor gene, involved in 13 q loss	V	NA		Genome instability and mutations; escaping programmed cell death
	*RB1*	RB transcriptional corepressor 1	13q14.2	Tumor suppressor gene, involved in 13 q loss	V	NA	Suppression of growth; escaping programmed cell death; change of cellular energetics	Escaping immunic response to cancer; invasion and metastasis; genome instability and mutations; escaping programmed cell death
	*GPC5*	Glypican 5	13q31.3	Tumor suppressor gene, involved in 13 q loss	V	NA	Suppression of growth; invasion and metastasis	
	*SOX21*	SRY-box transcription factor 21	13q32.1	Tumor suppressor gene, involved in 13 q loss	V	NA	Suppression of growth	Proliferative signaling
	*ERCC5*	ERCC excision repair 5, endonuclease	13q33.1	Tumor suppressor gene, involved in 13 q loss	V	NA	Genome instability and mutations; escaping programmed cell death	Genome instability and mutations
SP-MPE	*HOXB13*	Homeobox B13	17q21.32	Amplification	III	Tier II, level C	Change of cellular energetics	Escaping programmed cell death
SP-EPN-MYCN	*MYCN*	MYCN proto-oncogene, BHLH transcription factor	2p24.3	Amplification	II	Tier I, level A	Proliferative signaling; escaping immunic response to cancer; angiogenesis; genome instability and mutations; change of cellular energetics	Cell replicative immortality; invasion and metastasis; escaping programmed cell death

* The list of genes is selected from the Catalogue of Somatic Mutations in Cancer (COSMIC) Cancer Gene Census (https://cancer.sanger.ac.uk/census, accessed on 20 September 2021). Oncogenes and tumor suppressor genes are viewed as candidates for recurrent aberrations resulting in gain-of-function (1q gains) and loss-of-function (6q losses, 13q losses), respectively. ** Strength of evidence for gene diagnostic value based on Strength-of-evidence rating scheme of the Centre for Evidence-Based Medicine (https://www.cebm.net, accessed on 20 September 2021). *** Evidence-based variant (nucleotide substitution, copy-number variation, and fusion) of listed genes categorization based on the Joint Consensus Recommendation of the Association for Molecular Pathology, American Society of Clinical Oncology, and College of American Pathologists (AMP/ASCO/CAP recommendations). **** Potential roles of the cancer hallmark genes are annotated using COSMIC Cancer Gene Census (https://cancer.sanger.ac.uk/census, accessed on 20 September 2021), GeneCards: The Human Gene Database (https://www.genecards.org/, accessed on 20 September 2021), and KEGG: Kyoto Encyclopedia of Genes and Genomes (https://www.kegg.jp/kegg/, accessed on 20 September 2021). NA—nonapplicable.

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
