# Peer review of "Molecular Stratification of Childhood Ependymomas as a Basis for Personalized Diagnostics and Treatment"

_cancers, 2021, doi:10.3390/cancers13194954_

Round 1
Reviewer 1 Report
The resubmitted manuscript "Molecular Stratification of Childhood Ependymomas as a Basis for Personalized Diagnostics and Treatment" by Zaytseva et al. comprehensively describes the clasification of ependymomas.
The authors have succesfully adressed my previous concerns. Due to that I endorse the acceptance of the current manuscript.
Author Response
We thank the Reviewer for appreciating our manuscript.
Reviewer 2 Report
thanks for the response,
still the reading of the genes in each section seems redundant with no depth on the function of these genes.
please add a table summarizing the list of these genes and their implicated functions. I am not talking about the gene families.
the article in the current status reads like a clinical report!
Author Response
We thank the Reviewer for the constructive suggestion. We have added two new tables (Table 1 and Table 3 in revised manuscript) regarding subgroup-specific genes and candidate genes involved in recurrent chromosomal copy number aberrations, in addition to their implicated functions. We hope this information would help to clearly indicate hallmark of cancer in major molecular groups of ependymoma.
We highly appreciate the Reviewer's constructive and insightful comments, which have helped us to substantially improve our manuscript.
Reviewer 3 Report
Overall, the authors did a good job addressing the reviewers’ comments and revising the manuscript accordingly.
However, the authors are incorrect in saying on page 2 and 3 of the resubmission letter “we have focused on the molecular characteristics of ependymal tumors based on the recently published fifth edition of the WHO Classification of Tumors of the Central Nervous System” and on page 1, 2 and 12 of the manuscript “… in the newest 2021 WHO Classification of Tumors of the Central Nervous System” simply because the 2021 WHO reference book has not been published yet!! The IARC website (publications.iarc.fr) still lists the 2016 edition. Only a summary of the content of the upcoming new edition was published by editors of the WHO reference book (Ref. 9, Louis DN et al., 2021). In the absence of the final publication, the authors should not claim knowledge of the details of the molecular characteristics of ependymal tumors. Such information, when available, will help verify and provide additional detail of the molecular characteristics of ependymal tumors.
Author Response
We thank the Reviewer for pointing this out. We feel sorry for our mistake in presenting the data regarding the fifth edition of the WHO Classification of Tumors of the Central Nervous System. To be as clear as possible and in accordance with the Reviewer’s concerns, we have replaced “The newest 2021 classification of central nervous system (CNS) tumors is built on the integrated histo-molecular approach” by the phrase “The current trend in neuropathology directs to the integrated histo-molecular approach” and “The importance of integrated diagnosis for ependymomas is underscored in the newest fifth edition of the WHO Classification of Tumors of the Central Nervous System” by the phrase “The importance of integrated diagnosis for ependymomas is underscored in the recommendations of Consortium to Inform Molecular and Practical Approaches to CNS Tumor Taxonomy. These update recommendations were adopted and implemented by WHO experts” on page 1.
On page 2 phrase “This view has been supported by WHO experts and reflected in the most recent fifth edition of the WHO Classification of Tumors of the Central Nervous System (WHO CNS5)“ was replaced by “This view has been supported by WHO experts and reflected in the summary of upcoming fifth edition of the WHO Classification of Tumors of the Central Nervous System (WHO CNS5)”.
On page 15 of the revised manuscript the misleading phrase “This situation is stipulated in the newest 2021 WHO Classification of Tumors of the Central Nervous System“ was deleted.
We highly appreciate the Reviewer's insightful and helpful comments on our manuscript.
Round 2
Reviewer 2 Report
Accepted
Author Response

(The authors gave the same response as above.)

Reviewer 3 Report
Line 14: as the abstract refers to "molecular markers of decisive importance for the prognosis" of ependymomas, clearly distinguish clinical biomarkers from investigational biomarkers throughout the manuscript. Whenever available, include quantitative information in the text (e.g., prevalence, correlation value, p-value...) to support the evidence, and caution or consider removing markers of marginal value.
Line 34: change 'structure" to "prevalence".
Line 65: remove "Not Official".
Line 67: the changes made to the text "the summary of [the] upcoming fifth edition of the WHO Classification of Tumors of the Central Nervous System (WHO CNS5)" should be made consistently throughout the manuscript, i.e., at line 176, 345, 356 and 374, use the terms "the summary of the upcoming WHO CNS5".
Line 60: Table 2 "Summary of major molecular markers of ependymal tumor" should become Table 1.
Line 361: Table 3 "List of subgroup-specific recurrent diagnostic marker genes"? should become Table 2.
Line 305: Table 1 "List of candidate genes involved in recurrent chromosomal abnormalities in recurrent chromosomal abnormalities in PF-EFNs" should become Table 3. Indicate the strength of the evidence for each of these candidate genes.
Author Response
We thank the Reviewer for the constructive suggestions. The comments 2-4 have been corrected. As recommended by Reviewer, the appearance of the tables has been changed; Table 1 "List of candidate genes involved in recurrent chromosomal abnormalities in recurrent chromosomal abnormalities in PF-EFNs" and Table 3 "List of subgroup-specific recurrent diagnostic marker genes" have been merged. The strength of the evidence and evidence-based categorization of each particular gene have been indicated.
Pointing the prognostic value of EPN subgrouping, the body of the manuscript contains exclusively distinguished or discussed clinically relevant biomarkers. The prevalence of the recurrent genetic alterations has been reported in the manuscript; for the recently identified markers such as PLAGL1 rearrangements or ZFTA (non-RELA) fusions the prevalence remains unclear due to its rarity. Investigational biomarkers (candidate genes involved in recurrent chromosomal abnormalities) are summarized in the Table 2 and limited of the strength of the evidence is indicated.
This manuscript is a resubmission of an earlier submission. The following is a list of the peer review reports and author responses from that submission.
Round 1
Reviewer 1 Report
The manuscript by Zaytseva et al. "Molecular stratification of ependymomas as a basis for personalized diagnostics and treatment", provides a comprehensive review of the recent literature concerning ependymomas. I would like to conglatulate the authros for their thorough examination of the subject and the well-structured mini-review they prepared. The manuscript is well-written and easy to comprehend. My comments are as follows:
1) I think that the authors should add some additional comments on the molecular pathways that are active in each sub-type of ependymomas (1-2 sentences per sub-type).
2) I believe that a table summarizing genetic alterations in each sub-type, including the percentage of cases with the respective alteration, would be a nice addition.
Reviewer 2 Report
We read with interest the review article by Zaytseva et al discussing
Molecular Stratification of Ependymomas as new means for biomarkers and personalized medicine.
The article discusses the shortcomings of histopathological evaluation for reliable diagnostics, prognosis, and choice of treatment strategy. Thus, the review provides a comprehensive molecular-genetic characterization of ependymomas as a new means to stratify the different kinds of Ependymomas.
The authors discuss the genetic profiling of supratentorial (ST-EPNs), infratentorial (a.k.a. posterior fossa ependymomas, PF-EPNs), and spinal (Sp-EPNs) along with their by (epi)genetic features.
the overall reading of the review seems a bit redundant with no depth on the implications of the altered genetic profile and how these altered genes are implicated in each of these cancers. The work lacks any kind of mechanistic insights and is presented as a dry listing of genetic markers that are repetitive in each section.
This is not acceptable for a comprehensive review and won’t be of interest to readers in this area.
Secondly, the images and schematics are extremely basic with no reflection to the content or how these markers can be used as in personalized medicine.
This work has to be rewritten with more synthesis reflecting on how these genes are implicated in the pathogenesis of metastasis of and what are the molecular pathways involved….
I advise that the authors look at the studies by Michel Salzet et al where the studies indulge in genes/proteins and their different implications in cancer pathways.
Reviewer 3 Report
In the review article “Molecular stratification of ependymomas as a basis for personalized diagnostics and treatment”, the authors highlight the importance of the integrated diagnosis of ependymomas (EPNs) based on the combined histopathological and (epi)genetic profiling of these clinically and morphologically diverse tumors. This paradigm reflects the classification of EPNs in a recent summary (Ref. 8, Louis DN, et al. 2021) of the upcoming 2021 WHO Classification of Tumors of the Central Nervous System. EPNs are now classified by anatomic site [supratentorial (ST), posterior fossa (PF), spinal (SP)] as well as by histopathological (ependymoma, myxopapillary, subependymoma) and molecular features. The authors recognize molecularly defined types of ST-EPNs, those with RELA (defined by the presence of ZFTA-RELA fusions), with YAP1 (defined by the presence of YAP1-MAMLD1 fusions) and non-RELA/non-YAP1. There are also molecularly defined PF-EPNs, group PFA, group PFB and ST-RELA-like PF-EPNs. The SP-EPNs are defined by the presence of MYCN amplification. Myxopapillary EPNs and subependymoma remain tumor types because molecular studies have not provided additional classification criteria. The authors discuss laboratory approaches for EPN diagnosis.
Overall, the authors have put a very good effort reviewing and explaining the advances in the molecular classification of EPNs. Of particular utility is the detailed discussion of laboratory approaches for the diagnosis of EPNs. The molecular classification of EPNs has been the subject of several similar review articles, even recently. However, because the publication of the 2021 WHO Classification of Tumors of the Central Nervous System is imminent in the next few weeks, wouldn’t it be a good idea for the authors to incorporate in their manuscript information from that book once available and to be the first to write a review that includes the latest WHO classification of EPNs?
The authors should adhere to the nomenclature of EPNs described in Ref. 14 (Pajtler KW, et al., 2015) of the manuscript, namely ST-EPN-RELA, ST-EPN-YAP1, PFA, PFB, SP-EPN, etc. without modification to avoid confusion. On page 2 and 9, cite the appropriate references in the text after “in (recent/several) studies.” On page 4, make clear that PFA tumors are located more laterally in the PF and have a higher risk of recurrence and a worse prognosis than PFB tumors. In Figure 1, add the WHO grade. Under ST-RELA, add “chymotrypsis” after RELA fusions. Under ST-YAP1, indicate toddlers and children. On page 7, expand on the subject of myxopapillary EPNs and subependymoma and their clinicopathological features. On page 9, provide examples and references of the application of the NanoString nCounter platform to the study of EPNs. Before the conclusion, add a brief section that summarizes the current status of molecular targeting of EPNs using examples mentioned throughout the paper and additional references.
The list of references is not entirely up-to-date. If helpful for their review, the authors should consider recent articles such as Baroni LV, et al. Ultra high-risk PFA ependymoma is characterized by loss of chromosome 6q. Neuro Oncol. 2021 Aug 2;23(8):1360-1370. doi: 10.1093/neuonc/noab034; Jünger ST, et al. Pediatric ependymoma: an overview of a complex disease. Childs Nerv Syst. 2021 Aug;37(8):2451-2463. doi: 10.1007/s00381-021-05207-7; Łastowska M, et al. Transcriptional profiling of paediatric ependymomas identifies prognostically significant groups. J Pathol Clin Res. 2021 Jul 27. doi: 10.1002/cjp2.236; Mack SC, et al. Therapeutic targeting of ependymoma as informed by oncogenic enhancer profiling. Nature. 2018 Jan 4;553(7686):101-105. doi: 10.1038/nature25169.